# ChatGPT in Occupational Medicine: A Comparative Study with Human Experts

**DOI:** 10.3390/bioengineering11010057

**Published:** 2024-01-06

**Authors:** Martina Padovan, Bianca Cosci, Armando Petillo, Gianluca Nerli, Francesco Porciatti, Sergio Scarinci, Francesco Carlucci, Letizia Dell’Amico, Niccolò Meliani, Gabriele Necciari, Vincenzo Carmelo Lucisano, Riccardo Marino, Rudy Foddis, Alessandro Palla

**Affiliations:** 1Department of Translational Research and New Technologies in Medicine and Surgery, University of Pisa, 56126 Pisa, Italy; padovan.martina@gmail.com (M.P.); coscibianca@gmail.com (B.C.); armando.petillo@virgilio.it (A.P.); g.nerli@studenti.unipi.it (G.N.); f.porciatti@studenti.unipi.it (F.P.); scasers@gmail.com (S.S.); f.carlucci@studenti.unipi.it (F.C.); letizia.dellamico@gmail.com (L.D.); n.meliani1@studenti.unipi.it (N.M.); gabriele.necciari@gmail.com (G.N.); riccardo.marino@med.unipi.it (R.M.); 2Intel Corporation, Santa Clara, CA 95054, USA; alessandro.palla@intel.com

**Keywords:** artificial intelligence, ChatGPT, occupational health and safety, health promotion, digital health, large language model

## Abstract

The objective of this study is to evaluate ChatGPT’s accuracy and reliability in answering complex medical questions related to occupational health and explore the implications and limitations of AI in occupational health medicine. The study also provides recommendations for future research in this area and informs decision-makers about AI’s impact on healthcare. A group of physicians was enlisted to create a dataset of questions and answers on Italian occupational medicine legislation. The physicians were divided into two teams, and each team member was assigned a different subject area. ChatGPT was used to generate answers for each question, with/without legislative context. The two teams then evaluated human and AI-generated answers blind, with each group reviewing the other group’s work. Occupational physicians outperformed ChatGPT in generating accurate questions on a 5-point Likert score, while the answers provided by ChatGPT with access to legislative texts were comparable to those of professional doctors. Still, we found that users tend to prefer answers generated by humans, indicating that while ChatGPT is useful, users still value the opinions of occupational medicine professionals.

## 1. Introduction

Artificial intelligence (AI) has become an increasingly popular topic in healthcare, as proven by the rapidly increasing number of studies [1,2,3,4,5] and reviews [6,7,8,9] on this topic. More specifically, the use of AI chatbots like ChatGPT in healthcare is frequently discussed. ChatGPT is a publicly available chatbot created by OpenAI, a non-profit AI research and development company. It was released in November 2022 and can be accessed at chat.openai.com. AI chatbots typically consist of a chat interface and a large language model (LLM) based on the transformer architecture [10].

LLMs are powerful artificial neural networks trained on vast amounts of data to understand and generate natural languages. They can be used for translation, text creation, automated responses, and much more [11]. ChatGPT belongs to the Generative Pre-trained Transformer (GPT) [12] model topology and was fine-tuned from the GPT3.5 model using a massive corpus of chat conversations and trained on instruction-based following [13] and reinforcement learning from human feedback [14]. In March 2023, Open AI released GPT-4, an updated model of the GPT architecture [15]. At the time of our study, ChatGPT, as a standalone tool, did not have access to the internet but is now able to access a limited set of external data sources thanks to OpenAI plugins. To use ChatGPT, a prompt in a natural language is inputted, and the chatbot replies in the same language. The resulting experience closely resembles a real conversation between two individuals. Prompts are typically queries or instructions and can also incorporate diverse data inputs like research papers, mathematical equations, and spreadsheets [16,17].

Some healthcare-related uses of ChatGPT, including medical education, scientific research, medical writing, and diagnostic decision-making, are being explored [18,19]. More specifically, ChatGPT can generate scientific articles with appropriate vocabulary and a range of tones spanning from colloquial to highly technical [20]. ChatGPT could also be of valuable assistance to physicians in their clinical decision-making [21] by generating accurate lists of differential diagnoses [22] and providing insights for cancer screening decisions [23]. Integrating ChatGPT or using medical document templates can greatly streamline medical documentation, saving time and effort for healthcare professionals [24,25].

ChatGPT can be a valuable means of implementing an intelligent question–answering tool in the healthcare sector [19], as demonstrated by numerous studies evaluating the potential of GPT based chatbots in question–answer scenarios within specialized fields like microbiology [26], ophthalmology [27], radiology [28], and neurosurgery [29]. Most ChatGPT studies in medicine, while very promising, have only been tested through a multiple-choice simulated model. In contrast, the chatbot’s ability to provide healthcare professionals with more complex argumentative assistance remains much less investigated.

At the time of this analysis, there were no studies in the medical literature analyzing ChatGPT’s performance in the context of occupational medicine question–answering, while a study [30] already explored how it can provide a range of customized features to address challenges in the field of occupational health and safety. Our study aims to evaluate ChatGPT’s effectiveness in answering open-ended questions on the fundamental concepts of occupational medicine as established by general safety and occupational health principles and local sector regulations. Also, the study explores the potential implications and limitations of the use of ChatGPT in occupational health medicine and provides recommendations for future research.

## 2. Materials and Methods

The study involved a pool of twelve physicians, consisting of eight residents and four specialist physicians in occupational medicine, who were divided into two groups for blind review. In each group, there were two specialists and four residents. The work was supervised by a full professor of occupational medicine. 

### 2.1. Questions and Reference Answers Generation

The two groups of physicians were tasked with creating questions and their corresponding answers on the primary topics of occupational medicine. Each physician was assigned a specific topic and was instructed to refer to the most recent version of the legislative decree (D.lgs) 81/08, the Italian framework for occupational safety and health [31], for their questions. The topics covered the main hazards present in workplaces, including safety, biological, physical, ergonomic, chemical, and work organization hazards. The selected topics were workplace safety and use of work equipment, manual handling of loads, use of video terminals, physical hazards, chemical hazards, asbestos, carcinogenic and mutagenic agents, and biological and explosive atmospheres in the workplace. Each team member was tasked with generating approximately 40 questions and their respective answers, which were then uploaded on the dedicated Google Form platform. Overall, 433 questions and their respective reference answers were generated.

### 2.2. Bot-Generated Answers

We used the OpenAI ChatGPT API (Application Programming Interface) to reply to the physicians’ questions. The model used is gpt-3.5-turbo-0301, and the answers were generated using an automatic script on the 14th and 15th of April 2023 by interfacing directly with the OpenAI API. The method employed two distinct approaches. In the first approach, the questions generated by the two groups were fed directly into ChatGPT (an approach labeled ChatGPT). To mimic a real-use scenario, questions were submitted in Italian. 

ChatGPT’s system message serves as the primary instructions for the model, and it can be tailored to include various information about the system’s role. This can include a brief overview of the assistant, its personality traits, specific instructions or guidelines that you want the assistant to adhere to, or relevant data or information that the model should have, such as frequently asked questions. It is possible to personalize the system role to suit specific use cases. Although the system role/message is not mandatory, we found out that high-quality system messages improve the overall quality of the generated replies [32]. We perform prompt engineering [33] to optimize the system message on a small subset (10%) of the overall questions and reference answers and measure its effectiveness using automatic evaluation methods such as ROUGE [34] and LLM-based evaluation. We used automatic evaluation methods for prompt engineering to quickly iterate through different system configurations to find the most effective one. In addition, the prompt was fine-tuned to minimize hallucinations and non-relevant replies. More information about prompt engineering and implementation details can be found in Appendix A.

In the second approach, the reference legislative context from D.lgs 81/08 was also passed alongside the questions (approach labeled ChatGPT + context). To retrieve the context, we used semantic search with embeddings. Text embeddings are a technique employed in natural language processing (NLP) and machine learning that converts sentences into numerical vectors [35]. As text embeddings outperform traditional keyword-based searches and try to understand the meaning of the query and the context being searched, we used this technique to perform a semantic search to retrieve the relevant section from the D.lgs 81/80. The extracted legislative context was then passed alongside the original question to ChatGPT to generate the final reply. More information about this method and technical details can be found in Appendix A.

### 2.3. Answers Evaluation

The evaluation of generated responses involved a qualitative assessment of the accuracy, precision, completeness, usability, and relevance of each question and its three generated response options (ChatGPT, ChatGPT + context, and human answer generated during the question creation phase). Each question with its three response options was blindly evaluated by each physician, following pre-defined metrics. The evaluation criteria used to assess the quality of the generated responses are briefly described as follows:*Accuracy*: the answer provides correct and precise information without containing errors or inaccuracies;*Precision*: the answer is concise and to the point, without being overly verbose or ambiguous;Completeness: the answer provides all the necessary information to address the question;Usability: the answer is easy to understand and use for the target audience, which may consist of workers, managers, or safety professionals;Relevance: was the answer appropriate and relevant to the question asked.

In the evaluation phase, the user is prompted to choose the answer that performs better in each of the five metrics.

Additionally, a Likert 5-point scale was used to measure the absolute value of the accuracy and completeness of each generated response. This scale allowed for a more fine-grained evaluation of the quality of the responses by providing a numerical value to indicate the extent to which the response was accurate and complete. The Likert scale ranged from 1 (not accurate/complete at all) to 5 (completely accurate/complete), with intermediate values indicating varying degrees of accuracy and completeness.

This approach allowed for a more objective and quantitative evaluation of the quality of the generated responses, in addition to the qualitative evaluation based on accuracy, precision, usability, and relevance. The use of both qualitative and quantitative evaluation methods provided a comprehensive assessment of the quality of the generated responses and allowed for a more robust analysis of the results.

### 2.4. Evaluation Process and Criteria

To facilitate the evaluation process, a website was created where each physician could log in and access the questions and response options for evaluation. Information about the website used for the evaluation phase is presented in Appendix B. 

In the pursuit of maximizing comprehensiveness and impartiality in the evaluations, every physician from both groups scrutinized about 100 questions and their corresponding answers exclusively from the opposing group, with the evaluation sequence randomized across various users. This approach was implemented to minimize potential crosstalk between evaluations and ensure that all questions were covered as thoroughly as possible. For the same reason, the order of the answers (users’, ChatGPT, ChatGPT + context) is randomized in all evaluations to avoid any implicit bias.

### 2.5. Error Analysis in ChatGPT + Context Answers

After the evaluation phase, a questionnaire survey was conducted to investigate the specific errors encountered in the responses generated by ChatGPT + context. The questionnaire was administered alongside the poorly rated questions, which scored 2 or less on the Likert accuracy scale, to gain insights into the nature of these errors. 

The questionnaire also asked whether specific errors, such as typographical mistakes, grammatical errors, offensive or inappropriate language, internal contradictions, incorrect acronyms, or the misuse of Roman numerals, were present in the chatbot’s responses.

Additionally, participants were asked to assess the comprehensibility of the response and provide feedback if it was found to be unclear. Furthermore, respondents were encouraged to elaborate on their reasons for rating the question poorly. The questionnaire is available in Appendix C.

### 2.6. Error Analysis in ChatGPT + Context Answers

The statistical analysis was performed using Python 3.10 software and Pandas (version 2.0.0), NumPy (version 1.24.2), SciPy (version 1.10.1), Matplotlib (version 3.7.0), and Seaborn (version 0.12.2) libraries. Numerical values are expressed as averages and standard errors. The statistical differences between several groups were assessed using the two-sided Mann–Whitney U test. The statistical significance was accepted at *p* < 0.01. The source data is available upon request.

## 3. Results

Figure 1 shows the relative Likert score of ChatGPT, ChatGPT + context, and user-generated answers. Each set of questions, along with three alternative answers (users’, ChatGPT, ChatGPT + context), was evaluated by multiple physicians. We gathered a total of 1259 assessments, with each combination (question + answer) receiving an average of approximately 2.9 evaluations. In terms of accuracy, the users’ answers generally performed best, followed by the ChatGPT + context and the ChatGPT ones.

The accuracy scores for the responses generated by the physicians were significantly higher than those generated by the ChatGPT model, with an average score of 4.042 ± 0.032 for the physicians, 3.361 ± 0.035 for ChatGPT + context, and 3.091 ± 0.035 for the ChatGPT model (*p* < 0.001). 

The completeness scores for the responses generated by the physicians were significantly higher than those generated by the ChatGPT model, with an average score of 3.658 ± 0.030 for the physicians and 3.159 ± 0.033 for the ChatGPT model (*p* < 0.001). Although the completeness score was higher in physicians than in ChatGPT + context responses, the difference was not statistically significant (*p* = 0.862). All results are summarized in Table 1. Overall, the results suggest that the ChatGPT + context model can generate responses that are reasonably complete and accurate.

In terms of accuracy, ChatGPT garnered 17.5% preferences, ChatGPT + context received 34.8%, and users demonstrated the highest accuracy with 47.7% preferences. Similarly, regarding precision, ChatGPT achieved 18.0% preferences, ChatGPT + context had 37.4%, and users scored 44.6%. Completeness showed a similar trend, with ChatGPT at 18.2%, ChatGPT + context at 38.7%, and users at 43.1%. In terms of usability, ChatGPT obtained 16.6%, ChatGPT + context had 32.2%, and users showed the highest preference at 51.2%. Lastly, for relevance, ChatGPT received 18.7%, ChatGPT + context had 38.5%, and users scored 42.8%. Importantly, all differences were statistically significant across comparisons, with a *p*-value < 0.001 for all five metrics.

The results of the evaluation showed that ChatGPT + context responses were generally preferred to the ones generated from the model without context in all five metrics. However, users were found to have a stronger preference for human-generated responses in every metric. Results are shown in Figure 2. 

Following the evaluation, a questionnaire was dispatched to users to determine the nature of errors in ChatGPT + context responses. The questionnaire was sent together with the poorly rated questions (2 or less on the Likert accuracy scale). The results are depicted in Figure 3. The most common cause of error turned out to be wrong content (a wrong reply by the bot), followed by improper context (an off-topic response caused by a failure in the context passed to tpahe bot). Upon re-analyzing the generated responses, it was observed that, in most cases, the error originated from an incorrect reference context provided to ChatGPT by the semantic search process. We also found that ChatGPT quality degrades if the user provides a question with Roman numbers or acronyms, which are often mistaken for something else.

## 4. Discussion

The results obtained from this study represent the first evaluation of ChatGPT’s performance in generating responses to complex questions based on the regulatory context of Italian occupational medicine. In the study, it was found that the completeness of responses provided by ChatGPT with context is like that of responses given by doctors. However, when it comes to accuracy, although ChatGPT with context is better than the version without context, it does not reach the level of accuracy achieved by physicians. Even when comparing preferences for all five metrics, ChatGPT with context outperforms the version without context but falls short of the quality of responses generated by physicians, which were generally preferred by the evaluators. In our study, it is also noteworthy to observe that even human-generated responses did not consistently achieve a perfect score in terms of accuracy and comprehensiveness. This is because users can also make errors or may not have analyzed all the necessary information to respond accurately to the query. Also, the evaluation of human responses is often influenced by subjectivity and variability among different assessors. Evaluations may be based on personal opinions, cognitive biases, prior experiences, and individual interpretations. This can lead to variable scores in the assessment of human-generated responses as well.

In the literature, few studies have assessed ChatGPT’s performance in responding to open-ended medical questions. Johnson et al. [36] assessed the precision of cancer-related information delivered by ChatGPT when compared to the responses from the National Cancer Institute (NCI). They evaluated the accuracy (accurate: yes vs. no) and carried out a comparative analysis between the anonymized responses from NCI and ChatGPT. Their study indicates that ChatGPT delivers accurate information regarding common cancer-related information. In another study, Johnson D. et al. [37] assessed the accuracy and comprehensiveness of medical queries generated by healthcare professionals using ChatGPT. Subsequently, the physicians assigned ratings to the responses generated by ChatGPT based on their accuracy, measured on a 6-point Likert scale ranging from 1 (completely incorrect) to 6 (completely correct), and completeness, assessed on a 3-point Likert scale ranging from 1 (incomplete) to 3 (complete with additional context). The results indicated that across all the questions (in total, 284), the median accuracy score was 5.5 (indicating responses between almost completely and completely correct), with a mean score of 4.8 (signifying responses between mostly and almost completely correct). Furthermore, the median completeness score was three (reflecting complete and comprehensive responses), with a mean score of 2.5. Our study stands out as the first to test ChatGPT on workplace health and safety legislation. Furthermore, the metrics used in our study allow for an assessment not only of the accuracy and completeness of the chatbot-generated responses but also of the potential usability and relevance of these responses in the day-to-day practice of occupational health professionals. In addition, the errors found in responses that received low scores enabled the analysis of ChatGPT’s performance limitations. We also evaluated not only ChatGPT but also its integrated version with selected context, which demonstrates an increase in ChatGPT’s response performance.

Despite its promising performance in responding to open-ended questions, ChatGPT undoubtedly still has some important limitations. As ChatGPT training data goes only as far as the end of 2021, the chatbot cannot keep up with the latest trends and news, even though the introduction of plugins partially mitigates this time constraint. ChatGPT can also generate inconsistent or contradictory responses that do not align with medical guidelines [38]. In evaluating ChatGPT’s performance in responding to open-ended questions, the phenomenon of hallucinations must be taken into consideration. Hallucinations are instances where LLMs assertively incorporate incorrect details into the generated responses. Recognizing hallucinations is essential, but understanding how to mitigate their occurrence is equally crucial [39], while their mitigation remains an open question and is subject to numerous field studies [40,41]. However, it is a well-documented fact [42] that enhancing semantic research improves ChatGPT’s performance, something that aligns with the findings of our study, where the context-integrated version outperforms the one without context. Therefore, our future approach will continue to focus on refining the semantic research process to reduce text hallucinations. Additionally, through a survey sent to study participants, we tried to understand why the ChatGPT + context version received low accuracy scores in the answers to certain questions. It turned out that ChatGPT often misinterprets acronyms and Roman numerals and frequently makes references to a context that is unrelated to the question’s requirements. Consequently, by addressing these system limitations and coupling them with ongoing expert evaluations, we may enhance the quality of responses generated by ChatGPT. As a reference, we included some samples of responses from ChatGPT + context alongside the reference answers from users in Appendix D. There we reported an example of a reply evaluated highly by physicians and three hallucinated replies caused by a wrong reply (Table A4), an incorrect semantic search process (Table A5), and an incorrect identification of an acronym (Table A6).

There are also concerns about the ethical implications to consider regarding conversational AI in medical practice, as well as the potential for bias and errors in the data used to train AI algorithms. The legal implications of using these technologies cannot be downplayed. For instance, determining liability in the case of an inevitable mistake by an AI physician is yet to be established. The use of ChatGPT in medical practice can raise various legal and liability issues. If a medical error or negligence is attributed to the use of a system like ChatGPT, healthcare professionals and organizations may be held legally accountable [43]. Therefore, it is crucial to establish clear procedures and technology usage protocols.

Based on the findings of this study, several recommendations can be made for the future use of ChatGPT and LLMs in occupational health medicine. Firstly, it is important to ensure that the language model has full access to the legislative context to mitigate potential hallucinations and errors in its responses. ChatGPT accuracy greatly depends on the quality of the context data and the reliability of the semantic search process. This is particularly true in medicine, where LLMs’ errors and hallucinations may have potentially severe repercussions on people’s health. To mitigate these downsides, it is essential to ensure the use of accurate and up-to-date data from reliable sources (guidelines, regulations, etc.). However, because the study shows that even an occupational health LLM with full access to the legislative context is still prone to errors, human supervision is always necessary, and we do not suggest its direct and unfiltered interaction with patients who are unable to detect potential hallucinations. 

Additionally, ChatGPT should be used as a tool to support, rather than replace, human expertise and should not be currently relied upon as the sole source of information. ChatGPT cannot replace the experience, knowledge, and skills of a professional medical doctor in the diagnosis and treatment of medical conditions, as well as in establishing a doctor-patient relationship. Human medical professionals play an indispensable role in the decision-making process. 

Furthermore, it is important to implement appropriate safeguards to protect the privacy and security of user data when using ChatGPT, especially if the model is deployed in the cloud and patients’ information is sent to a third party. Open-source LLMs like Llama [44], Alpaca [45], and Vicuna [46] are now approaching the level of quality of proprietary models like ChatGPT and do not need to send data in the cloud, as they run fully on the user’s PC and thus eliminate any privacy concerns. However, these models require system memory and computational power well beyond the reach of the average user’s laptop. Nevertheless, next-generation Central Processing Units (CPU) from all major manufacturers will have an AI accelerator, so running an LLM locally on a user’s laptop will become accessible to everyone.

The use of ChatGPT and similar AI can bring numerous benefits to the field, including round-the-clock assistance to occupational physicians, increased efficiency, cost reduction, health monitoring, and personalized services. With ChatGPT, occupational physicians could ask questions about workplace safety, workers’ rights, safety protocols, and more at any time and receive immediate responses. Its impact on health and safety, as well as societal values, requires critical reflection. However, based on the ethical and legal limitations and implications, a responsible, ethical, and reflective approach is necessary for their development and use. It is important that developers, practitioners, and policymakers work together to ensure that the use of such tools is ethical, transparent, and accountable and that appropriate safeguards are in place to protect both users and professionals in the field of occupational health.

Several limitations of this study can be identified that we plan to overcome in future work. One possible limitation of this work is the small sample size of employed physicians and the fact that this is a monocentric study. Moreover, questions were created solely based on legislative decree 81/08, which represents the reference legislation in the field of occupational medicine. However, there are also numerous other laws or decrees that regulate the protection of workers’ health and safety in the workplace, for example, legislative decree n. 66/2003 about night work or legislative decree n.151/2001 on the protection of working mothers.

## 5. Conclusions

Our study revealed that ChatGPT + context outperformed ChatGPT in responding to complex open-ended questions based on the regulatory context of Italian occupational medicine, even though it did not reach the level of accuracy and completeness of the questions generated by physicians. This study not only underscores ChatGPT’s potential as a valuable resource within the field of occupational health but also emphasizes the need to acknowledge the current limits of AI. 

To assess chatbots’ potential in occupational medicine as virtual assistants for medical inquiries in workplace health and safety, we plan to improve both the quality of the reference questions and answers and their size by employing a more diverse, multi-center, and wider sample of occupational health professionals, including medical residents, academics, occupational physicians, and other professionals working in the field of workplace health and safety.

Our team also aims to conduct a follow-up study to compare the performance of the new version of ChatGPT with GPT-4 and Google’s Bard AI. Further studies should be encouraged to explore other possible applications of ChatGPT and other large language models in occupational medicine.

## Figures and Tables

**Figure 1 bioengineering-11-00057-f001:**
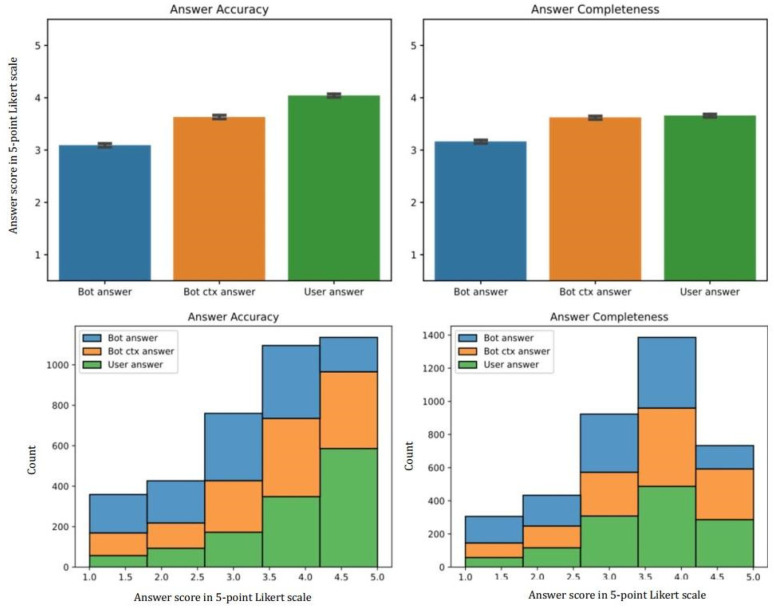
Accuracy and completeness Likert score and distribution for ChatGPT, ChatGPT + context, and users’ generated questions.

**Figure 2 bioengineering-11-00057-f002:**
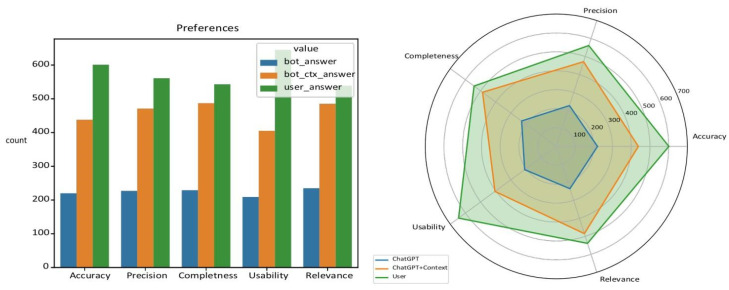
Results of the evaluation of the generated response options (ChatGPT, ChatGPT + context, human answer) for each question in terms of accuracy, precision, completeness, usability, and relevance.

**Figure 3 bioengineering-11-00057-f003:**
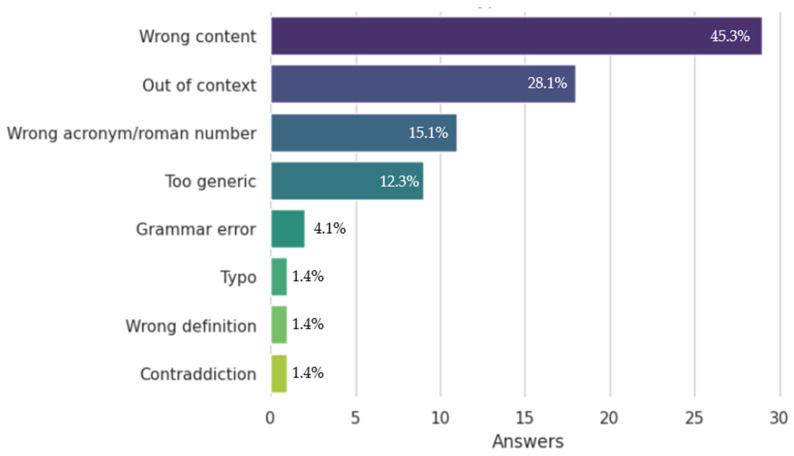
The results of the questionnaire were conducted to investigate the most frequent specific errors encountered in the answers generated by ChatGPT + context.

**Table 1 bioengineering-11-00057-t001:** Comparative scores and *p*-values for Physicians’, ChatGPT, and ChatGPT + context.

	Physicians vs. ChatGPT		Physicians vs. ChatGPT + Ctx		ChatGPT vs. ChatGPT + Ctx	
Metric	Values	*p*-Value	Values	*p*-Value	Values	*p*-Value
Compl.	3.658 vs. 3.159	<0.001	3.658 vs. 3.618	0.862	3.159 vs. 3.618	<0.001
Accuracy	4.042 vs. 3.091	<0.001	4.042 vs. 3.631	<0.001	3.091 vs. 3.631	<0.001

## Data Availability

The data presented in this study are available on request from the corresponding author. The data are not publicly available as their release is subject to ongoing collaboration and further analysis to derive additional insights.

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
