# Peer review of "ChatGPT in Occupational Medicine: A Comparative Study with Human Experts"

_bioengineering, 2024, doi:10.3390/bioengineering11010057_

Round 1

Reviewer 1 Report

Comments and Suggestions for Authors

This well written paper covers a very important and timely topic.  It is critical that it shows the accuracy of GPT open access and how it can be improved using context addition. The area of occupational medicine chosen is particularly important since the information provided needs to be accessible and understandable to non-medical personnel. The methodology followed with the two groups, the blind review, and the rating is excellent.

Also the Appendices included provide the reader with the relevant information and, in particular, the examples of hallucinations are spot on in explaining the problems encountered.

I look forward to reading the follow-up study for ChatGPT 4!

Comments on the Quality of English Language

The paper needs editing by a native speaker because there are some minor grammatical errors making some parts difficult to follow. 

Author Response

Dear Reviewer 1,

We sincerely appreciate the time and effort you dedicated to reviewing our paper titled "ChatGPT in Occupational Medicine: A Comparative Study with Human Experts” Regarding your observation about grammatical errors, we acknowledge the importance of clear and concise communication. Thus, we have engaged the services of a linguistic expert for a thorough review and correction of grammatical issues. If you have any additional insights or recommendations, please feel free to share them with us. We value your input and look forward to the possibility of working with Bioengineering again in the future. Thank you for your time and dedication to advancing the quality of academic research.

Best Regards,

Dr. Martina Padovan

Reviewer 2 Report

Comments and Suggestions for Authors

The paper tries to measure how chatGPT can be supportive in coping with challenging questions in the field of occupational medicine. The goal is achieved by comparing the answers with domain expert contribute and additional domain-related documents.

In general, ChatGPT seems to be a promising tool, even if still not ready to be used for daily practice.

The English can (should) be improved and matherial and methods is not clearly documented. The experimental setting has significant limitations, such as: (i) it is limited to the italian language (ii) it is a mono-centric study with a significant risk of bias (iii) the adopted version of ChatGPT is going to be obsolete 

However, it seems to be a communicative, reasonable (even if a bit "naive") attempt to take stock of the current, promising, LMM-based technologies.

some suggestions:

ABSTRACT:

>> "The objective of this study is to evaluate ChatGPT’s accuracy and reliability in answering complex medical questions related to occupational health and to explore the implications and limitations of AI in occupational health medicine while providing recommendations for future research in this area and informing decision-makers about AI impact in healthcare."

I'd suggest to split this LONG sentence by punctuation.

INTRODUCTION

>> "LLM are powerful artificial neural networks trained on vast amounts of data from the internet to generate text and understand natural language"

not necessarly via internet. They only need texts, internet is only one of the possibility to find the text.

>> "At the time we performed this analysis, there were no studies in the medical literature that analyze ChatGPT's performance in occupational medicine"

Please, consider now to update the paper with:

Sridi C, Brigui S. The use of ChatGPT in occupational medicine: opportunities and threats. Ann Occup Environ Med. 2023 Oct 23;35:e42. doi: 10.35371/aoem.2023.35.e42. PMID: 38029273; PMCID: PMC10654530.

MATHERIAL AND METHODS

>> "The study involved a pool of physicians, including of resident and specialist physicians in occupational medicine, who were divided into two groups for blind review"

How many, exactly, in total? How many residents and specialists per group?

MATHERIAL AND METHODS (2.1)

>> "D.lgs 81/08"

Please, explain the meaning of this contracted expression

MATHERIAL AND METHODS (2.2)

>> "We used OpenAI ChatGPT API to reply to the physicians’ questions"

please, specify VERSION, release, etc. and the precise time range (date of the first question - date of the last question). In addition: did you use APIs or your worked with a direct input by chat?

>> "In the second approach, the reference legislative context from the D.lgs 81/08 was also passed alongside the questions [..]"

This second approach, starting with the aforementioned sentence, is interesting but not clear. I'd ask you to explain IN DETAIL how it works, step by step this approach, also including a figure, if possible.

(e.g.: what does "the reference legislative context from 399 the D.lgs 81/08 was also passed alongside the questions" mean? Was the D.lgs 81/08 used to train the LLM in some way? I don't understand)

MATHERIAL AND METHODS (2.2)

>> "Each question with its three response options was evaluated by each physician who was blinded to the group that had generated the question"

It sounds twisted, like an Italian sentence converted by ChatGPT.

MATHERIAL AND METHODS (2.3)

>> "Accuracy: The answer prov [..]"

I appreciate your idea and your performance index. Well thought. However, because "accuracy" and "precision" have a different meaning, in statistics, I'd suggest to use, at least, those terms in italic.

APPENDIX 1

Appendix A.2 ChatGPT + Context generated answers

>> "In simpler terms, text embeddings are a way of representing words and paragraphs as numbers, which can be more easily analyzed and processed by  computers"

mmmhhh.... was this sentence written by a computer scientist?

>>  'UnstructuredPDF' library

Fine but in which version? Which sw environment?

In general, in this section, the method sounds a bit unclear, poorely commented and the English limited or, at least, not technical.

Comments on the Quality of English Language

Limited English: long and convoluted sentences. Although accurate, some parts of the text seem to be written by authors not always competent in the vocabulary of computer science

Author Response

Dear Reviewer 2,

Thank you for your insightful and constructive feedback on our paper titled "ChatGPT in Occupational Medicine: A Comparative Study with Human Experts.” We appreciate the time and effort you have invested in reviewing our work.

We also appreciate your valuable observations regarding the need for improvement in the English language, as well as the documentation of materials and methods. We addressed these aspects thoroughly in the revised version of the paper. Thus, we have engaged the services of a linguistic expert for a thorough review and correction of grammatical issues.

Regarding the limitations you pointed out, we understand your concerns about the experimental setting. We acknowledge that the study is limited to the Italian language, and it is indeed a mono-centric study with potential bias. We also take note of your mention that the adopted version of ChatGPT may become obsolete. We opted for ChatGPT version 3.5 because was freely accessible to everyone at the time of the study (our analysis phase was conducted in April), and the availability and accessibility of ChatGPT 3.5 at that time influenced our decision, since at that time GPT4 API wasn’t fully accessible to everyone. We are planning to publish a GPT4 version after this paper get published.

We are aware of the methodological limitations in our study, but we want to emphasize that our intention for this work to serve as a starting point to promote the dissemination of artificial intelligence, specifically Large Language Models (LLM), in the field of occupational medicine on an international scale. Despite its central role in safeguarding the health and safety of workers, occupational medicine is often overlooked compared to other medical disciplines. For this reason, we hope that, beyond the acknowledged limitations, you can appreciate the aim of our work, which is to contribute to bringing greater attention and visibility to this field on the international stage.

Our motivation to submit our work to a peer-reviewed journal was further bolstered upon learning about the special issue Applications of Large Language Models in Medical and Biomedical Data Processing in Bioengineering. We believe that our study contributes valuable insights to the intersection of artificial intelligence and occupational medicine, and we hope it will find resonance within the bioengineering community.

Your feedback is crucial in enhancing the quality and rigor of our research, and we are committed to making the necessary improvements. If you have any specific suggestions or further recommendations, please do not hesitate to share them with us.

Once again, we appreciate your time and effort in reviewing our paper, and we look forward to the possibility of addressing your concerns in the revised manuscript.

Following the point-by-point response to your feedback

ABSTRACT:

>> "The objective of this study is to evaluate ChatGPT’s accuracy and reliability in answering complex medical questions related to occupational health and to explore the implications and limitations of AI in occupational health medicine while providing recommendations for future research in this area and informing decision-makers about AI impact in healthcare." I'd suggest to split this LONG sentence by punctuation.

As suggested, we have split the long sentence in the abstract into two sentences by punctuation.

INTRODUCTION

 >> "LLM are powerful artificial neural networks trained on vast amounts of data from the internet to generate text and understand natural language" not necessarily via internet. They only need texts, internet is only one of the possibility to find the text.

To avoid confusion, we have removed 'from the internet,' even though the data used to train the LLMs comes from massive data scraping from the internet. As an example, see Figure 2.2 in “Language model are Few-Shot Learners” (https://arxiv.org/abs/2005.14165) citation 11 in the paper, that represent the original GPT3 paper from witch GPT3.5 and ChatGPT were trained using SFT and RLHF. There the composition of the dataset was fully disclosed and consists mostly of data crawled from the web as well as few books’ datasets. The training datasets for new models are not fully disclosed as OpenAI main competitive advantage.

>> "At the time we performed this analysis, there were no studies in the medical literature that analyze ChatGPT's performance in occupational medicine"Please, consider now to update the paper with:Sridi C, Brigui S. The use of ChatGPT in occupational medicine: opportunities and threats. Ann Occup Environ Med. 2023 Oct 23;35:e42. doi: 10.35371/aoem.2023.35.e42. PMID: 38029273; PMCID: PMC10654530.

Thank you for pointing out the paper “The use of ChatGPT in occupational medicine: opportunities and threats”. We have included the citation in the introduction to contextualize our work within the landscape of existing research. Additionally, we have specified that our study is currently the only one evaluating the performance of ChatGPT in occupational medicine to provide appropriate context for readers and highlight the contribution of our research. 

MATHERIAL AND METHODS

 "The study involved a pool of physicians, including of resident and specialist physicians in occupational medicine, who were divided into two groups for blind review" How many, exactly, in total? How many residents and specialists per group?

We specified the number of participants in the text, as well as the breakdown between residents and specialists. 

MATHERIAL AND METHODS (2.1)

>> "D.lgs 81/08"

Please, explain the meaning of this contracted expression

We explained the meaning of the contracted expression D.lgs 81/08. 

MATHERIAL AND METHODS (2.2)

>> "We used OpenAI ChatGPT API to reply to the physicians’ questions" please, specify VERSION, release, etc. and the precise time range (date of the first question - date of the last question). In addition: did you use APIs or your worked with a direct input by chat?

In the text, we stated that the model used is gpt-3.5-turbo-0301, and the answers were generated using an automated script on the 14th and 15th of April 2023 by interfacing directly with the OpenAI API. This allowed us to generate all the replies very quickly since the entire process (semantic search, query generation, system message construction, API query and answers storage) was made fully automatic using a python script.

>> "In the second approach, the reference legislative context from the D.lgs 81/08 was also passed alongside the questions [..]" This second approach, starting with the aforementioned sentence, is interesting but not clear. I'd ask you to explain IN DETAIL how it works, step by step this approach, also including a figure, if possible (e.g.: what does "the reference legislative context from 399 the D.lgs 81/08 was also passed alongside the questions" mean? Was the D.lgs 81/08 used to train the LLM in some way? I don't understand)

The details about how the second approach are explained in Appendix A.2, A.3 and A4, that has been expanded to be more technical especially about what embeddings are that we agree was too semplicistic and not in line with a journal like bioengeneering that has broad technical audience. Appendix A.2 explains the semantic search process, what are word embeddings and how we parsed the 81/08 document to create a vector database for retrivial augmented generation. Appendix A.3 details how we use the retreived legislative context alongside the LLM system message. Please note that Table 3 contains a full example of the a system message we used and the {context} part refers the the legislative context extracted by the semantic search part as explained in the caption. Appendix A.4 explains how we use prompt engineering to finetune the system hyperparameters by using ChatGPT itself as a few-shot evaluator. Table 4 shows the evaluator system message we use to pass the question and reference answer. Furthermore we added Figure 4 that fully summarize how the vector database was constructed and how ChatGPT + Context works. Training ChatGPT wasn’t a possibility at the time we did this study, as it was made available only on August (https://openai.com/blog/gpt-3-5-turbo-fine-tuning-and-api-updates).

While we agree that the technical appendix was lacking of details and we took action to improve it, we are a little bit reluctant to put too many details as the methodology deserves a paper on its own, especially the parts about retrivial augmented generation (method 2 => ChatGPT + Context) and the prompt engineering optimization. We beleive that the techniques we used might be better discussed in a standalone paper, possibly not limited to Occupational health, published alongside a high quality software packet for building this kind of system for a broader range of medical fields.

MATHERIAL AND METHODS (2.2)

>> "Each question with its three response options was evaluated by each physician who was blinded to the group that had generated the question" It sounds twisted, like an Italian sentence converted by ChatGPT.

We have revised the sentence that sounded twisted.

MATHERIAL AND METHODS (2.3)

>> "Accuracy: The answer prov [..]" I appreciate your idea and your performance index. Well thought. However, because "accuracy" and "precision" have a different meaning, in statistics, I'd suggest to use, at least, those terms in italic.

Thank you for acknowledging the concept and the performance index I will incorporate this recommendation into the document.

APPENDIX 1

Appendix A.2 ChatGPT + Context generated answers

>> "In simpler terms, text embeddings are a way of representing words and paragraphs as numbers, which can be more easily analyzed and processed by computers" mmmhhh.... was this sentence written by a computer scientist?

We improve the aforementioned section. Further details about what we have done are in the response to material and methods.

>> ‘UnstructuredPDF' library Fine but in which version? Which sw environment?In general, in this section, the method sounds a bit unclear, poorely commented and the English limited or, at least, not technical.

We added the version of the library we used. Since it was requested, we added also the version of every software packet we used for this analysis in section 2.3.